# Structural basis of epilepsy-related ligand–receptor complex LGI1–ADAM22

Atsushi Yamagata[1,2,3], Yuri Miyazaki[4,5], Norihiko Yokoi [4,5], Hideki Shigematsu[6], Yusuke Sato[1,2,3], Sakurako Goto-Ito[1,2], Asami Maeda[1,2], Teppei Goto[7], Makoto Sanbo[7], Masumi Hirabayashi[5,7], Mikako Shirouzu[6], Yuko Fukata[4,5], Masaki Fukata[4,5] & Shuya Fukai [1,2,3]

Epilepsy is a common brain disorder throughout history. Epilepsy-related ligand–receptor complex, LGI1–ADAM22, regulates synaptic transmission and has emerged as a determinant of brain excitability, as their mutations and acquired LGI1 autoantibodies cause epileptic disorders in human. Here, we report the crystal structure of human LGI1–ADAM22 complex, revealing a 2:2 heterotetrameric assembly. The hydrophobic pocket of the C-terminal epitempin-repeat (EPTP) domain of LGI1 binds to the metalloprotease-like domain of ADAM22. The N-terminal leucine-rich repeat and EPTP domains of LGI1 mediate the intermolecular LGI1–LGI1 interaction. A pathogenic R474Q mutation of LGI1, which does not exceptionally affect either the secretion or the ADAM22 binding, is located in the LGI1–LGI1 interface and disrupts the higher-order assembly of the LGI1–ADAM22 complex in vitro and in a mouse model for familial epilepsy. These studies support the notion that the LGI1–ADAM22 complex functions as the trans-synaptic machinery for precise synaptic transmission.

[1] Institute for Quantitative Biosciences, The University of Tokyo, Tokyo 113-0032, Japan. [2] Synchrotron Radiation Research Organization, The University of Tokyo, Tokyo 113-0032, Japan. [3] Department of Computational Biology and Medical Sciences, Graduate School of Frontier Sciences, The University of Tokyo, Chiba 277-8561, Japan. [4] Division of Membrane Physiology, Department of Molecular and Cellular Physiology, National Institute for Physiological Sciences, National Institutes of Natural Sciences, Okazaki 444-8787, Japan. [5] Department of Physiological Sciences, School of Life Science, SOKENDAI (The Graduate University for Advanced Studies), Okazaki 444-8787, Japan. [6] RIKEN Center for Life Science Technologies, Tsurumi-ku, Yokohama 230-0045, Japan. [7] Center for Genetic Analysis of Behavior, National Institute for Physiological Sciences, National Institutes of Natural Sciences, Okazaki 444-8787, Japan. These authors contributed equally: Atsushi Yamagata, Yuri Miyazaki. Correspondence and requests for materials should be addressed to M.F. (email: mfukata@nips.ac.jp) or to S.F. (email: fukai@iam.u-tokyo.ac.jp)

Epilepsy is one of the most common neurological disorders, which affects around 1% of the population. Epilepsy is featured by recurrent, unprovoked seizures, which are caused by an imbalance between excitation and inhibition in neural circuits. Epilepsy-related mutations often occur in genes of ion channels regulating neuronal excitability, such as voltage-gated ion channels ($K^+$, $Na^+$, and $Ca^{2+}$) and ligand-gated ion channels (nicotinic acetylcholine and $GABA_A$ receptors)[1–3]. Some other epilepsy-related mutations have been found in genes encoding non-ion channel proteins such as *LGI1*.

LGI1 is a 60-kDa secreted neuronal protein, which consists of the N-terminal leucine-rich repeat (LRR) domain and the C-terminal epitempin-repeat (EPTP) (also known as EAR) domain[4] (Fig. 1a). Mutations of *LGI1* cause autosomal dominant lateral temporal lobe epilepsy (ADLTE; also known as autosomal dominant partial epilepsy with auditory features (ADPEAF))[5–7]. To date, at least 42 *LGI1* mutations have been reported in ADLTE families, including 28 missense mutations that are distributed in both the LRR and EPTP domains (Supplementary Table 1)[5,6,8–29]. Most of the ADLTE missense mutations are secretion-defective, suggesting that they affect folding and/or posttranslational modifications of LGI1. Actually, a secretion-defective E383A mutant of LGI1 is recognized by the endoplasmic reticulum (ER) quality control machinery and prematurely degraded to cause epilepsy in a mouse model of ADLTE[9]. In addition to *LGI1* mutations in inherited epilepsy, autoantibodies against LGI1 most frequently occur with limbic encephalitis (LE) presenting with acquired amnesia and seizures in adults[30–32].

ADAM22 is a member of transmembrane ADAM metallo-proteases but is catalytically inactive. ADAM22 serves as a receptor for LGI1 and is anchored to the excitatory postsynaptic density through PSD-95 scaffold[33]. The LGI1–ADAM22 ligand–receptor interaction plays an essential role in AMPA-type glutamate receptor-mediated synaptic transmission via PSD-95[33–35]. A global LGI1 protein complex determined by proteomic analysis contains ADAM22 subfamily members (ADAM22, ADAM23, and ADAM11) as LGI1 receptors, postsynaptic scaffold proteins (PSD-95, PSD-93, and SAP97), and also presynaptic potassium channels (Kv1) and scaffolds (CASK and Lin7)[34,36]. Genetic evidence that loss of *Lgi1*[34,37,38], *Adam22*[39], *Adam23*[40], or *Kv1* channels[41,42] in mice causes a similar lethal epileptic phenotype supports their actions in a linear molecular pathway. Importantly, reported *LGI1* mutations[9], *ADAM22* mutations in a patient with seizures and intellectual disability[43], and LGI1 autoantibodies in patients with LE[32] all converge on the disruption of the LGI1–ADAM22 ligand–receptor interaction. Thus, LGI1–ADAM22 interaction is essential for physiological brain excitability and functions.

LGI1 might serve as the ligand that tethers ADAM22 and ADAM23 at the synaptic cleft and trans-synaptically couple postsynaptic AMPA receptors on the PSD-95 platform with presynaptic machinery containing potassium channels[34,36]. However, structural mechanisms underlying this tethering model remain elusive, due to the lack of three-dimensional (3D) structural information of LGI1 and its complex with the ADAM22 subfamily proteins. In this study, we present the crystal structures of LGI1 LRR, LGI1 EPTP–ADAM22, and LGI1–ADAM22 at 1.78, 2.67, and 7.13 Å resolutions, respectively. Together with the structure-guided functional studies, we reveal the structural basis for pathogenesis of epilepsy that is associated with the trans-synaptic interaction mediated by the higher-order assembly of LGI1–ADAM22 subfamily proteins.

## Results

### Structure of LGI1 EPTP–ADAM22 ectodomain complex.
The C-terminal EPTP domain of LGI1 is sufficient for binding to the ectodomain (ECD) of ADAM22[33] (Fig. 1a). We crystallized the complex between LGI1 EPTP and ADAM22 ECD to elucidate the mechanism of the interaction between LGI1 and ADAM22. The expression level of LGI1 EPTP alone in Expi293F cells was too low for crystallization. Co-expression with ADAM22 ECD was required to obtain a sufficient amount of LGI1 EPTP. The crystal structure of the LGI1 EPTP–ADAM22 ECD complex was determined at 2.67 Å resolution by molecular replacement using the ADAM22 ECD structure[44] (PDB 3G5C) as the search model (Fig. 1b and Table 1). LGI1 EPTP folds into a seven-bladed β-propeller (blades 1–7) (Supplementary Fig. 1a). Each blade is composed of a four-stranded antiparallel β-sheet (strands A–D) (Supplementary Fig. 1b). The N-terminal strand is assembled with the C-terminal three strands to form blade 7. The disulfide bond between Cys260 in blade 1 and Cys286 in blade 2 stabilizes the whole β-propeller structure. Inside the central channel, $Ca^{2+}$ is coordinated by the side chains of Asp334, Glu336, and Asp381, the main-chain O atom of Val382, and a water molecule bound to Glu383 (Supplementary Fig. 1c). The EPTP domain is structurally related to a WD40 domain. A structure-based sequence alignment of the blades in LGI1 EPTP unveils WD-like sequence motifs within the blades (Supplementary Fig. 1d, e). The position of the WD-like motif in LGI1 EPTP is shifted by two residues from those in other canonical WD40 proteins, indicating that LGI1 EPTP is an atypical WD40 domain.

ADAM22 ECD consists of four domains (Fig. 1a, b): a metalloprotease-like domain (residues 233–435), a disintegrin domain (residues 445–529), a cysteine-rich domain (residues 530–676), and an epidermal growth factor (EGF)-like domain (residues 677–718)[44]. The metalloprotease-like domain of ADAM22 binds to the EPTP domain of LGI1 with a buried surface area of 1034 $Å^2$ (Fig. 1b). Trp398, Tyr408, and Tyr409 of ADAM22 are stacked in layer and project into the inner rim of the central channel of LGI1 EPTP to interact hydrophobically with Phe256, Val284, Leu302, Tyr433, and Met477 of LGI1 (Fig. 1c). In addition, four hydrogen bonds are formed between LGI1 and ADAM22: Arg330 and Lys331 of LGI1 hydrogen bond with the side and main chains of Asp405 of ADAM22, respectively, and Lys353 and Arg378 of LGI1 hydrogen bond with Glu359 of ADAM22. In our pull-down analyses, the W398D, Y408A, Y409A, or Y408A Y409A mutation of ADAM22 almost or completely abolished its binding to LGI1, indicating that the hydrophobic interaction mediated by Trp398, Tyr408, and Tyr409 of ADAM22 is essential for binding between LGI1 and ADAM22 (Fig. 1c–e). On the other hand, the E359A or D405A mutation of ADAM22 decreased but did not abolish its binding to LGI1. The hydrogen bonds play a secondary role in binding between LGI1 and ADAM22 (Fig. 1c–e). More quantitative molecular interaction analysis such as surface-plasmon resonance spectroscopy, isothermal titration calorimetry, or other comparable biophysical techniques was not applicable, owing to extreme difficulty in preparing LGI1 alone.

The apo structure of ADAM22 ECD[44] (PDB 3G5C) is essentially the same as the LGI1-bound structure (Cα rmsd of 0.9 Å), except for the Trp398- and Tyr408–Tyr409-containing loops (Fig. 1f). These aromatic residues are buried inside the protein in the apo state and become exposed to LGI1 EPTP upon binding. Mechanistically, Lys331 and Arg378 of LGI1 appear to eject the side chains of Tyr409 and Tyr408 of ADAM22, respectively. The disulfide bond between Cys394 and Cys401 tethers the N- and C-terminal ends of the Trp398-containing loop to support its conformational change (Fig. 1f). The C401Y mutation of ADAM22 impairs the binding to LGI1 in vitro[43] and has been found in a patient with rapidly progressing severe encephalopathy with intractable seizures and profound

intellectual disability[43]. Trp398, Cys394, and Cys401 are completely conserved in ADAM11, ADAM22, and ADAM23 (Fig. 1g).

**LGI1 EPTP-binding modes of ADAM22 and ADAM23.** ADAM23, another LGI1 receptor, likely interacts with LGI1 in a manner similar to ADAM22, since the amino-acid sequence identity between ADAM22 and ADAM23 is substantially high (e.g., ~50% between human ADAM22 and ADAM23). To support this idea, we mutated ADAM22-interacting residues of LGI1 and compared the effects of the mutations on the binding to ADAM22 and ADAM23 by pull-down assays (Fig. 2).

The Y433A and M477A mutations of LGI1, which disturb the hydrophobic interaction with Tyr408 of ADAM22, almost abolished the binding to both ADAM22 and ADAM23 (Fig. 2). Tyr408 of ADAM22 is replaced by Val in ADAM23 (Fig. 1g), which may also hydrophobically interact with Tyr433 and Met477 of LGI1. The F256A, V284A, and L302A mutations of

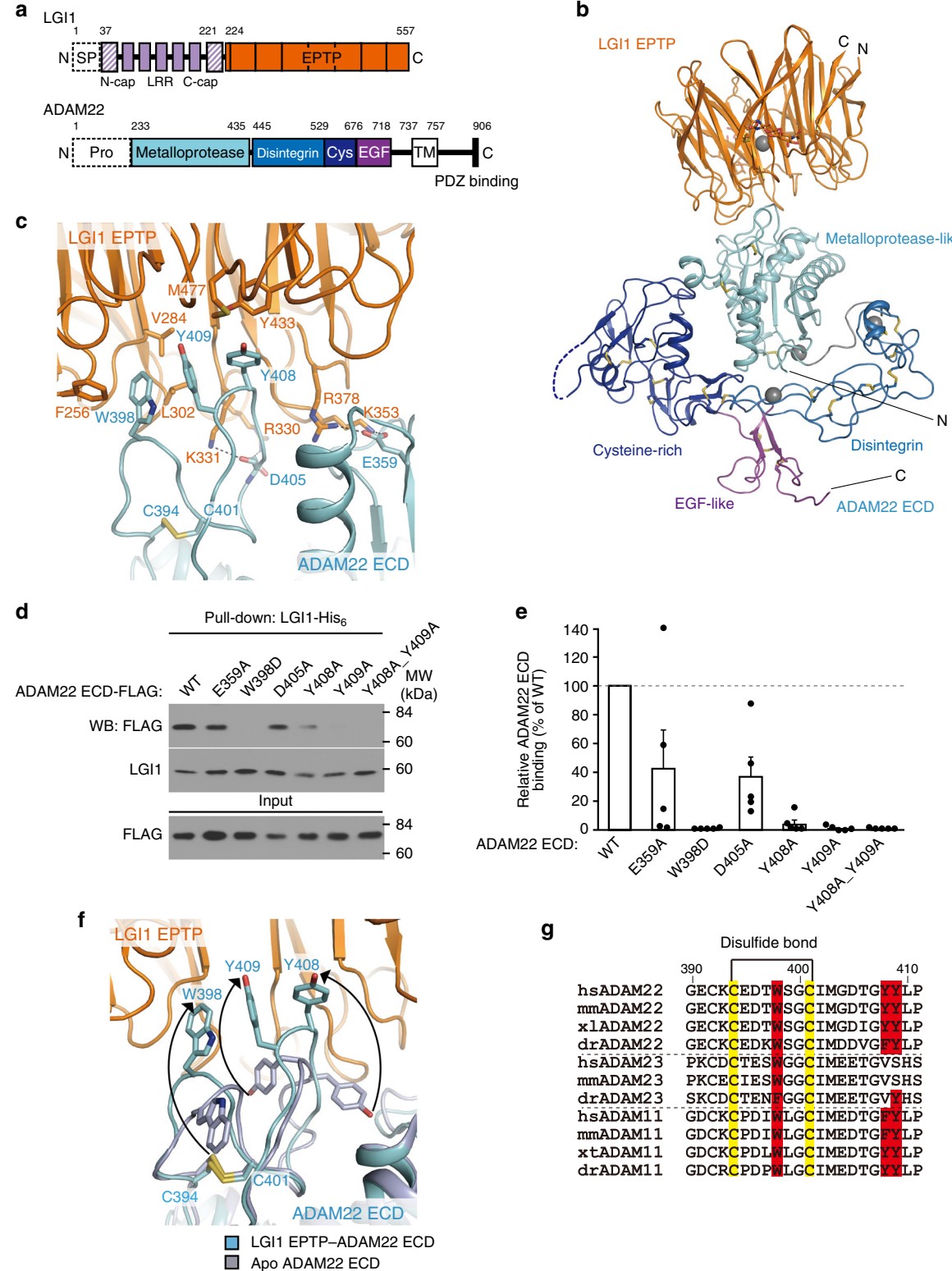

LGI1, which disturb the hydrophobic interaction with Trp398 of ADAM22, drastically impaired the binding to both ADAM22 and ADAM23 (Fig. 2). Greater effects of the F256A and V284A mutations of LGI1 on the binding to ADAM23 than on the binding to ADAM22 seem to be related to the difference in their hydrophobic interactions with Tyr433 and Met477 of LGI1; the replacement of Tyr408 in ADAM22 by Val in ADAM23 (Fig. 1g) may decrease the total affinity of ADAM23 to LGI1. The R330A and K331A mutations of LGI1, which disable the hydrogen bonding with Asp405 of ADAM22, modestly impaired the binding to both ADAM22 and ADAM23 (Fig. 2). The R330A K331A double mutation almost abolished the binding (Fig. 2). Similarly, the K353A and R378A mutations of LGI1, which disable the hydrogen bonding with Glu359 of ADAM22, substantially impaired or almost abolished the binding to both ADAM22 and ADAM23 (Fig. 2). As mentioned above, Arg378 of LGI1 appears to function in the displacement of Tyr408 of ADAM22 upon binding to LGI1. A greater effect of the R378A mutation on the binding to ADAM23 than on the binding to ADAM22 suggests that the displacement of Tyr408 of ADAM22 occurs more easily than that of the corresponding Val of ADAM23, along with the stacking interaction with Tyr409 (Fig. 1c, f), which is replaced by Ser in ADAM23 (Fig. 1g).

Taken together, these results suggest that LGI1 binds to ADAM22 and ADAM23 in a similar manner. As such, one LGI1 molecule cannot bind simultaneously to both ADAM22 and ADAM23 through the EPTP domain; at least two LGI1 molecules should be required for the suggested trans-synaptic tethering between ADAM22 and ADAM23[34].

**Mapping of ADLTE mutations on LGI1 structure.** To date, 42 *LGI1* mutations have been reported in patients with familial ADLTE[9,10,13,14,24]. To gain structural insights into pathogenic mechanisms of these mutations, we mapped 28 missense mutations onto the LGI1 structure (Fig. 3a, b). For this mapping, we also determined the crystal structure of LGI1 LRR alone at 1.78 Å resolution (Fig. 3b and Table 1). Nineteen of the examined missense mutations are secretion-defective, likely owing to failure of protein folding[9,26]. Correspondingly, the C42R, C42G, C46R, C46F, C179R, and C200R mutations disrupt intramolecular disulfide bonds in the N- and C-terminal caps of LGI1 LRR (Fig. 3b), which are common in extracellular and membrane-associated LRR proteins to stabilize their N- and C-terminal edges[45]. The E383A mutation disables the water-mediated $Ca^{2+}$ coordination that stabilizes the β-propeller structure of LGI1 EPTP (Fig. 3b and Supplementary Fig. 1c). Other secretion-defective mutations affect structural cores inside LGI1 LRR or EPTP (Fig. 3b).

Three secretion-competent mutations, R407C, S473L, and R474Q, are located in LGI1 EPTP (Fig. 3a, b)[9,27,28,46]. The S473L mutation specifically impairs the binding to ADAM22 in vivo[9], although Ser473 is located distant from the ADAM22-interacting region (Supplementary Fig. 2a). One possible mechanism for this

### Table 1 Data collection and refinement statistics

|  | LGI1 EPTP–ADAM22 ECD | LGI1 LRR | LGI1–ADAM22 ECD |
|---|---|---|---|
| **Data collection** | | | |
| Wavelength (Å) | 1.0000 | 1.0000 | 1.0000 |
| Resolution (Å) | 50.0–2.67 (2.72–2.67) | 50.0–1.78 (1.81–1.78) | 50.0–7.12 (7.24–7.12) |
| Space group | P1 | P6₁ | P2₁ |
| Cell dimensions | | | |
| $a, b, c$ (Å) | 83.6, 83.6, 293.5 | 65.3, 65.3, 109.7 | 105.1, 124.3, 164.7 |
| $\alpha, \beta, \gamma$ (°) | 86.4, 88.2, 59.7 | 90.0, 90.0, 120.0 | 90.0, 104.8, 90.0 |
| Completeness (%) | 90.6 (88.4) | 99.9 (99.8) | 97.9 (96.3) |
| $CC_{1/2}$ | (0.702) | (0.556) | (0.578) |
| $R_{sym}$ (%) | 12.1 (53.2) | 5.5 (45.4) | 9.9 (46.1) |
| $I/\sigma I$ | 9.9 (1.6) | 47.2 (2.2) | 13.3 (1.8) |
| Redundancy | 3.8 (3.7) | 15.1 (8.4) | 7.1 (4.6) |
| **Refinement** | | | |
| Resolution (Å) | 48.8–2.67 | 39.4–1.78 | 49.0–7.13 |
| No. reflections | 177,245 | 25,295 | 6086 |
| No. atoms | | | |
| Protein | 35,375 | 1460 | 15,684 |
| Sugar | 376 | – | 140 |
| Ion | 24 | – | 8 |
| Water | 914 | 100 | – |
| $R_{work}/R_{free}$ (%) | 24.4/27.9 | 16.6/19.1 | 26.5/31.6 |
| R.m.s.d. | | | |
| Bond lengths (Å) | 0.005 | 0.008 | 0.004 |
| Bond angles (°) | 0.777 | 1.103 | 0.819 |
| *Average B (Å²)* | | | |
| Protein | 46.0 | 44.4 | 448.1 |
| Sugar | 73.4 | – | 421.7 |
| Ion | 35.3 | – | 442.1 |
| Water | 37.6 | 54.1 | – |
| Ramachandran plot | | | |
| Most favored (%) | 96.6 | 97.9 | 96.0 |
| Disallowed (%) | 0.0 | 0.0 | 0.12 |

Values in parentheses are for the highest-resolution shell

**Fig. 1** Structure of LGI1 EPTP–ADAM22 ECD complex. **a** Domain organizations of LGI1 and ADAM22. LGI1 consists of the LRR (purple) and EPTP (orange) domains. The N-terminal secretion signal peptide (SP, enclosed by dotted lines) is removed in the secreted LGI1. The shaded purple boxes represent the N- and C-terminal caps, whereas the filled purple boxes represent the LRRs. The orange boxes represent the blades of the β-propeller. The premature form of ADAM22 contains the N-terminal prosequence (enclosed by dotted lines). The mature ADAM22 consists of the metalloprotease-like (cyan), disintegrin (light blue), cysteine-rich (dark blue), EGF-like (purple), transmembrane (white), and cytoplasmic domains. The major ADAM22 isoform has a PDZ-binding motif in the C-terminal region of the cytoplasmic domain. **b** Overall structure of LGI1 EPTP–ADAM22 ECD complex. The bound calcium ions are shown as gray spheres. The *N*-linked sugar chains and disulfide bonds are shown as sticks. The coloring scheme is the same as that in **a**. **c** Close-up view of the interface between LGI1 EPTP and ADAM22 ECD. The residues involved in their binding and a disulfide bond between Cys394 and Cys401 of ADAM22 are shown as sticks. Hydrogen bonds are shown as dotted lines. The coloring scheme is the same as that in **a**. **d, e** Pull-down assay between LGI1-His₆ and ADAM22 ECD-FLAG mutants. LGI1-His₆ and the indicated ADAM22 mutant proteins secreted from HEK293T cells were mixed and pulled-down with Ni-NTA agarose. Shown are Western blots (WB) of the (co-)purified (upper two panels) and input (bottom) samples with indicated antibodies (**d**). Quantification of the amounts of the co-purified ADAM22 ECD mutant proteins with LGI1 is shown in the graph (**e**). Results are shown as mean ± s.e. (*n* = 5 independent experiments). **f** Conformational change in Trp398, Tyr408, and Tyr409 of ADAM22 upon binding to LGI1 EPTP. The apo-ADAM22 structure (light purple, PDB 3G5C) is superposed onto the LGI1 EPTP–ADAM22 ECD structure (cyan). **g** Amino-acid sequence alignment of ADAM22, ADAM23, and ADAM11 from representative vertebrates (hs *Homo sapiens*, mm *Mus musculus*, xl *Xenopus laevis*, xt *Xenopus tropicalis*, dr *Danio rerio*), generated by ClustalW[70]

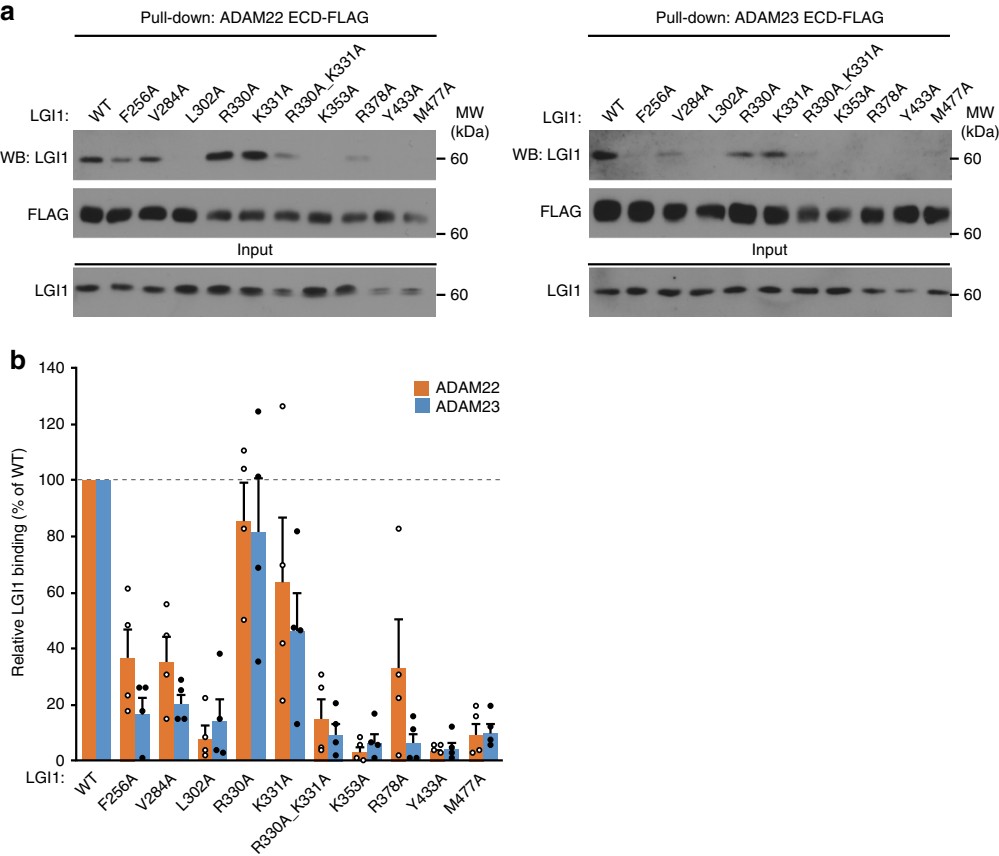

**Fig. 2** Site-directed mutational analysis of the interaction between LGI1 and ADAM22 or ADAM23. Pull-down assay between LGI1 mutants and ADAM22 ECD or ADAM23 ECD. Indicated LGI1 mutants and ADAM22 ECD-FLAG or ADAM23 ECD-FLAG secreted from HEK293T cells were mixed and pulled-down with anti-FLAG antibody agarose. Shown are Western blots of the (co-)purified (upper two panels) and input (bottom) samples with indicated antibodies (**a**). Quantification of the amounts of the co-purified LGI1 with ADAM22 ECD-FLAG (orange) or with ADAM23 ECD-FLAG (light blue) is shown in the graph (**b**). Results are shown as mean ± s.e. ($n = 4$ independent experiments)

ADAM22-specific impairment of the binding by the S473L mutation is that Phe451 of LGI1, which is located in the vicinity of Ser473 (<5Å), may shift and cause a steric hindrance with Tyr408 and/or Lys362 of ADAM22 (Supplementary Fig. 2a). In the present LGI1 EPTP–ADAM22 ECD structure, Phe451 of LGI1 is 5–6 Å apart from Tyr408 and/or Lys362 of ADAM22, which are replaced by amino-acid residues of shorter side chains in ADAM23 and ADAM11 (Val/Leu and Phe/Asn, respectively) to avoid steric hindrance with Phe451 of LGI1. In contrast to S473L, neither the R407C nor R474Q mutation reduced the binding to ADAM22 and ADAM23 in vitro (Supplementary Fig. 2b and Fig. 3c, respectively). Both Arg407 and Arg474 of LGI1 are exposed to the solvent and not involved in any interactions in the complex between LGI1 EPTP and ADAM22 ECD (Fig. 3b). The structure of the LGI1 EPTP–ADAM22 ECD complex provided little information about the structural mechanism for pathogenesis of either the R407C or R474Q mutation. Consistent with the functional and structural features of the LGI1 R407C mutant (LGI1[R407C]; superscripts attached with protein names hereafter denote their mutations), homozygous null Lgi1[−/−] mice, which showed spontaneous recurrent generalized seizures and premature death, could be rescued by the reexpression of Lgi1[R407C] transgene (Lgi1[−/−];R407C) in the brain, similar to the reexpression of Lgi1[WT] transgene (Lgi1[−/−];WT)[34] (Fig. 3d). Furthermore, when the expressed LGI1[R407C] was purified from the mouse brain, LGI1[R407C] bound to ADAM22 and ADAM23 as LGI1[WT] did (Supplementary Fig. 2c). Consistently, LGI1[R407C] variant was found in five gnomAD controls (the Genome Aggregation Database; http://gnomad. broadinstitute.org/). We thus conclude that R407C is not a pathogenic mutation for ADLTE. In striking contrast, the reexpression of Lgi1[R474Q] transgene (Lgi1[−/−];R474Q) could not rescue the epileptic phenotype of the Lgi1[−/−] mouse, showing the premature death due to lethal epilepsy (Fig. 3d). As the expression level of LGI1[R474Q] protein in the Lgi1[−/−];R474Q mutant mouse was similar to that of LGI1[WT] protein (Fig. 3d), the R474Q mutation is pathogenic to impair the LGI1 function other than the binding to the ADAM22 family proteins, without affecting the folding of LGI1. The information on the ADLTE-associated mutations described here was summarized as Supplementary Table 1.

**Defect in higher-order LGI1–ADAM22 assembly by LGI1[R474Q].** The molar mass of the LGI1 EPTP–ADAM22 ECD complex determined by size-exclusion chromatography coupled with multi-angle laser light scattering (SEC-MALS) was 108 kDa under our standard condition (i.e., in the presence of 150 mM NaCl), which is consistent with the theoretical molar mass of the 1:1 complex (95 kDa; without sugar chains) (Fig. 4 and Table 2). On the other hand, under the same condition, the determined molar mass of the complex between the full-length LGI1 (hereafter referred to as LGI1[WT] when compared with mutant LGI1) and ADAM22 ECD (356 kDa) was about three times larger than the theoretical molar mass of the 1:1 complex (117 kDa; without sugar chains) (Fig. 4 and Table 2), suggesting the higher-order assembly of LGI1–ADAM22. We hypothesized that this assembly of LGI1–ADAM22 ECD might reflect the trans-synaptic assembly

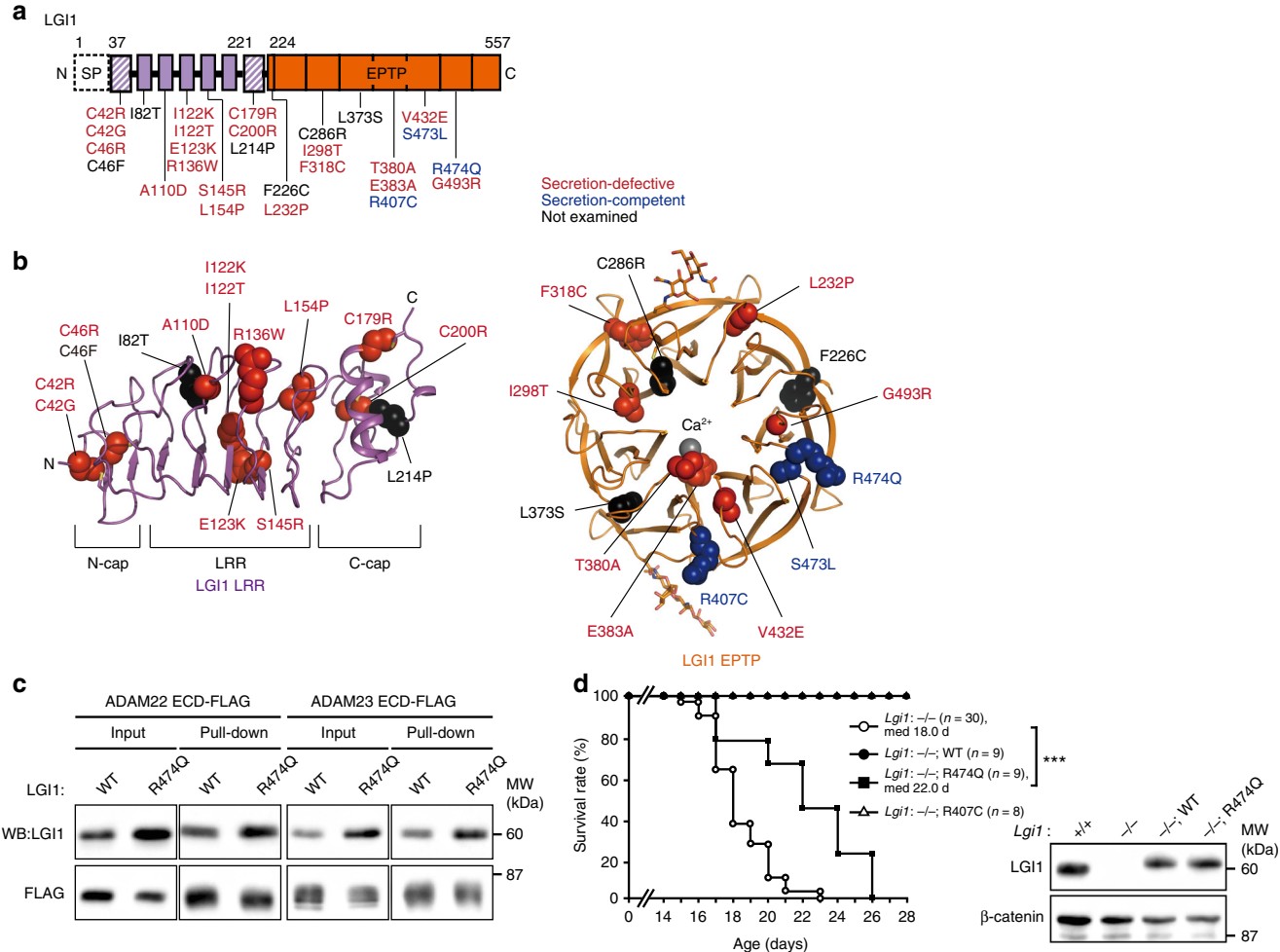

**Fig. 3** Missense ADLTE mutations in LGI1. **a** Schematic representation of 28 missense ADLTE mutations in LGI1 (red, secretion-defective; blue, secretion-competent; black, not examined). The drawing scheme of the domain organization of LGI1 is the same as that in Fig. 1a. **b** Mapping of 28 missense ADLTE mutations in LGI1 LRR and EPTP on the LGI1 structure. The amino-acid residues in the mutation sites are shown as spheres. The coloring scheme of the mutation sites is the same as that in **a**. **c** Pull-down assay between LGI1$^{WT}$ or LGI1$^{R474Q}$ and ADAM22 ECD-FLAG or ADAM23 ECD-FLAG. Shown are Western blots of the (co-)purified (pulled-down with FLAG antibody agarose) and input samples with indicated antibodies. **d** Kaplan–Meier survival plots of mice with secretion-competent ADLTE mutations. ***$P < 0.001$, log-rank test. Loss of LGI1 in $Lgi1^{-/-}$ mice and reexpression of LGI1 in $Lgi1^{-/-;WT}$ mice and $Lgi1^{-/-;R474Q}$ mice were confirmed by Western blots of brain lysates with indicated antibodies

of LGI1 with ADAM22 and ADAM23 in the brain and that LGI1$^{R474Q}$–ADAM22 lacks this assembly activity. To test this hypothesis, we first examined the effect of the R474Q mutation of LGI1 on the higher-order LGI1–ADAM22 ECD assembly. The molar mass of the LGI1$^{R474Q}$–ADAM22 ECD complex was analyzed by SEC-MALS and compared with those of the LGI1$^{WT}$–ADAM22 ECD and LGI1 EPTP–ADAM22 ECD complexes (Fig. 4 and Table 2). By co-expressing with ADAM22 ECD, LGI1$^{R474Q}$ was successfully prepared as the complex with ADAM22 ECD as well as LGI1$^{WT}$ or LGI1 EPTP. The determined molar mass of the LGI1$^{R474Q}$–ADAM22 ECD complex was 134 kDa, which corresponds to the theoretical molar mass of the 1:1 complex (117 kDa; without sugar chains), indicating that the R474Q mutation of LGI1 prevents the higher-order assembly of LGI1–ADAM22 ECD in vitro.

**Dimer-of-dimer assembly of LGI1–ADAM22.** The molar mass of LGI1$^{WT}$–ADAM22 ECD determined under our standard condition (356 kDa) suggested a trimer-of-dimer assembly of LGI1–ADAM22 ECD (Fig. 4 and Table 2). However, we suspected that the trimer-of-dimer assembly seems unlikely because

LGI1 LRR and LGI1 EPTP–ADAM22 themselves exhibit no obvious structural features suggestive of trimer formation (Figs. 1b and 3b). We then assessed the buffer condition for SEC-MALS and found that the determined molar mass of the LGI1$^{WT}$–ADAM22 ECD complex can be changed in a manner dependent on NaCl concentration, in contrast to those of the LGI1 EPTP–ADAM22 ECD and LGI1$^{R474Q}$–ADAM22 ECD complexes (Fig. 4 and Table 2). The determined molar mass of LGI1$^{WT}$–ADAM22 ECD at 500 mM NaCl was 267 kDa, corresponding to the 2:2 tetrameric assembly of LGI1–ADAM22 ECD.

To clarify the stoichiometry and assembly mode of LGI1–ADAM22 ECD, we crystallized the complex consisting of the full-length LGI1 and ADAM22 ECD. The complex between the full-length LGI1 and ADAM22 ECD was prepared by co-expressing LGI1 and ADAM22 ECD in a similar manner to the LGI1 EPTP–ADAM22 ECD complex. We coincidentally found that the R470A mutation of LGI1 enhances the expression level of the complex. LGI1$^{R470A}$ did not reduce the binding to ADAM22 or ADAM23 (Supplementary Fig. 3a). The molar mass of the LGI1$^{R470A}$–ADAM22 ECD complex determined by SEC-MALS was nearly identical to that of the LGI1$^{WT}$–ADAM22 ECD complex (Supplementary Fig. 3b). We therefore used the

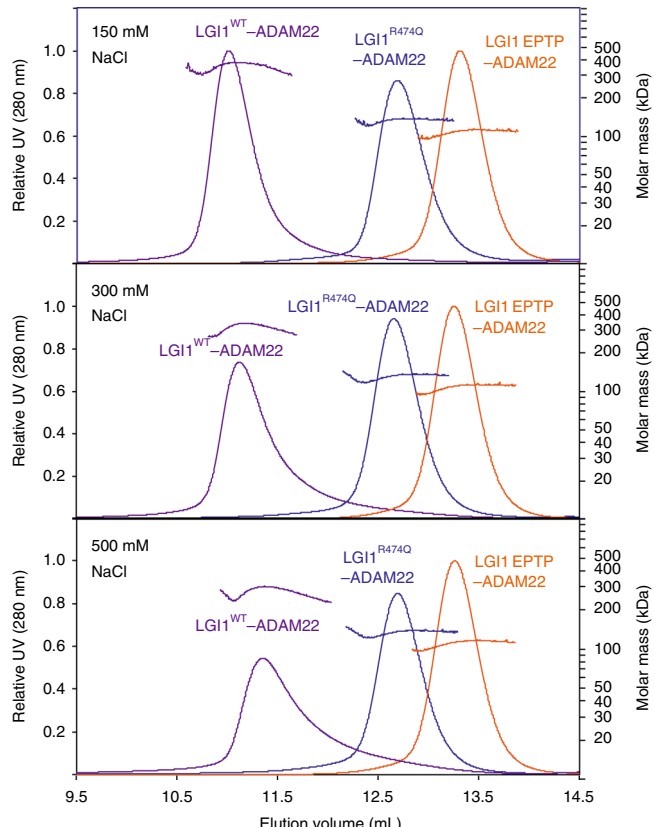

**Fig. 4** SEC-MALS analysis of LGI1–ADAM22 ECD complex. SEC-MALS analyses of LGI1WT–ADAM22 ECD (purple), LGI1R474Q–ADAM22 ECD (blue), and LGI1 EPTP–ADAM22 ECD (orange) in 20 mM Tris-HCl (pH 7.5) buffer containing 150 mM, 300 mM, or 500 mM NaCl. Chromatograms and determined molar masses are shown

LGI1R470A–ADAM22 ECD complex for crystallization. The crystal structure of the LGI1R470A–ADAM22 ECD complex was determined at 7.13 Å resolution by molecular replacement using the crystal structures of the LGI1 EPTP–ADAM22 ECD complex and LGI1 LRR as the search models (Fig. 5a and Table 1). As expected, Arg470 of LGI1 (replaced by Ala in the present structure) seems not to be involved in the LGI1–ADAM22 interaction (Supplementary Fig. 3c). The LGI1R470A–ADAM22 ECD complex is hereafter referred to as the LGI1–ADAM22 complex unless otherwise noted. The LGI1–ADAM22 complex structure forms a 2:2 heterotetramer in the asymmetric unit of the crystal (Fig. 5a). The length along the longest axis of the 2:2 LGI1–ADAM22 complex is about 190 Å, which is equivalent to the length of a synaptic cleft. Two copies of the 1:1 LGI1–ADAM22 complex are aligned in a head-to-head configuration with ~90° rotation along the longest axis. The LRR and EPTP domains of LGI1 are connected by a 2-residue linker (Ile222–Ile223) and adopt an extended conformation. LGI1 EPTP interacts with the metalloprotease-like domain of ADAM22 in the same manner as in the LGI1 EPTP–ADAM22 ECD complex, whereas the LRR domain of one LGI1 molecule interacts with the EPTP domain of the other LGI1 molecule, thereby bridging two distant ADAM22 molecules in the complex. The C-terminals of the two ADAM22 molecules are oriented in the opposite directions. No interactions between the two LGI1 LRRs were observed in the 2:2 LGI1–ADAM22 complex.

Cryo-electron microscopy (cryo-EM) of the LGI1R470A–ADAM22 ECD complex supported the presence of the 2:2 LGI1–ADAM22 complex in solution (Supplementary Fig. 4). Even though the sample was purified and analyzed to be in

**Table 2 Summary of SEC-MALS analysis of LGI1–ADAM22 ECD complex**

| LGI1-ADAM22 | Determined molar mass (kDa) | | | Theoretical molar mass (kDa) |
|---|---|---|---|---|
| | [NaCl] (mM) | | | |
| | 150 | 300 | 500 | |
| LGI1WT–ADAM22 | 356 | 320 | 267 | 234 (2:2 complex) |
| LGI1R474Q–ADAM22 | 134 | 131 | 129 | 117 (1:1 complex) |
| LGI1 EPTP–ADAM22 | 108 | 108 | 106 | 95 (1:1 complex) |

higher-order assembly (i.e., trimer-of-dimer or dimer-of-dimer assembly), the micrographs showed monomer, dimer, and oligomer of particles in the vitreous ice. Reference-free two-dimensional (2D) classification of more than 70,000 particles from 750 micrographs to 250 classes showed classes of monomer, dimer, and trimer of about 95 Å diameter particles, which are supposed to be the LGI1–ADAM22 complex. The numbers of the particles corresponding to classes of the 1:1, 2:2, and 3:3 LGI1–ADAM22 complexes are found to be 46,153 (65%), 21,163 (30%), and 3780 (5%), respectively. Most of the classes in the 2:2 complex resemble the calculated projection images from the crystal structure of the 2:2 LGI1–ADAM22 complex. Many of these classes showed clear averaged images of the 1:1 complex in one particle but blurred the other particle (likely to be another 1:1 complex), suggesting that the dimer assembly of the LGI1–ADAM22 complex is flexible (Supplementary Fig. 4). The 3:3 LGI1–ADAM22 complex seems to be less flexible than the 2:2 complex, especially around the boundaries between the adjacent particles (Supplementary Fig. 4).

The high-resolution structures of the LGI1 EPTP–ADAM22 ECD complex and LGI1 LRR allowed us to interpret the 7.13-Å-resolution map and obtain the information of the intermolecular interactions between LRR and EPTP and that between LRR and ADAM22 ECD in the 2:2 heterotetrameric LGI1–ADAM22 complex (Fig. 5b, c): Glu123 and Arg76 in the LRR domain of one LGI1 likely form hydrogen bonds with Arg474 and Glu516 in the EPTP domain of the other LGI1, respectively. His116 of LGI1 LRR likely interacts with Glu446 in the disintegrin domain of ADAM22. It should be noted that two human ADLTE mutations, E123K[17] and aforementioned R474Q[28], occur in the interface of the LGI1–LGI1 interaction in the 2:2 LGI1–ADAM22 complex (Fig. 5b). Although LGI1E123K secretion was heavily disturbed, LGI1R474Q protein was normally secreted from transfected HEK293T cells[9] and bound to ADAM22 and ADAM23 as LGI1WT (Fig. 3c). Given that the R474Q mutation of LGI1 is actually pathogenic to cause epilepsy in mice (Fig. 3d), the structure of the 2:2 LGI1–ADAM22 complex supports the notion that the pathogenic mechanism of LGI1R474Q is a defect in the assembly of the heterotetrameric ADAM22–LGI1–LGI1–ADAM22/23 complex in synapses. The LGI1–LGI1 interface found in the 2:2 LGI1–ADAM22 complex was further validated by SEC-MALS analysis of the LGI1R76A–ADAM22 ECD complex, which also showed the disruption of the higher-order assembly (Supplementary Fig. 3b).

**Disruption of LGI1–LGI1 interaction causes epilepsy**. To prove the R474Q mutation-mediated pathogenic mechanism, we then tandem-affinity purified the LGI1 protein complexes from mouse brains in which LGI1WT or LGI1R474Q tagged with FLAG and His6 was reexpressed in the *Lgi1* knockout background. Similar band patterns were obtained between LGI1WT- and LGI1R474Q-

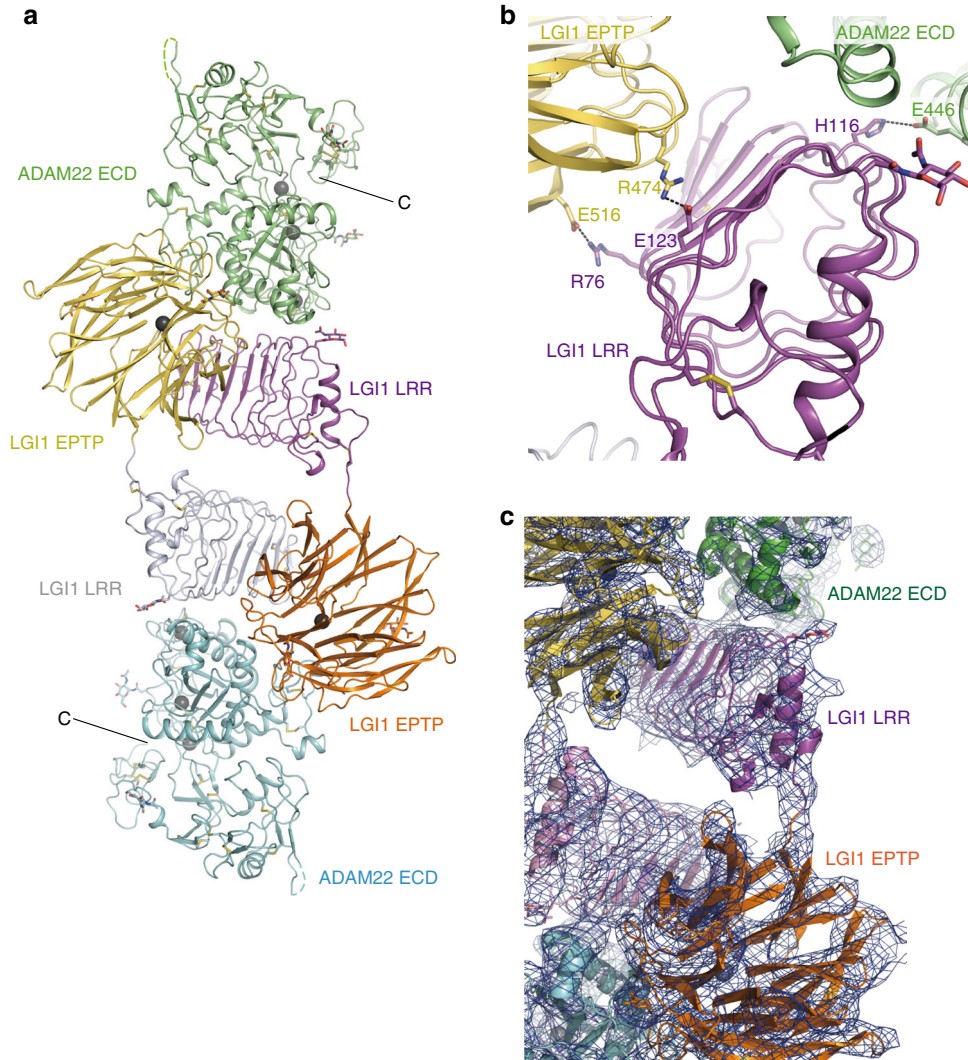

**Fig. 5** Overall architecture of the 2:2 LGI1–ADAM22 complex. **a** Overall structure of the 2:2 LGI1–ADAM22 complex. LGI1 LRR, LGI1 EPTP, and ADAM22 ECD in one LGI1–ADAM22 pair are colored in magenta, orange, and cyan, respectively, whereas those in the other pair are colored in gray, light yellow, and light green, respectively. **b** Close-up view of the LGI1 LRR–LGI1 EPTP and LGI1 LRR–ADAM22 ECD interfaces in the 2:2 LGI1–ADAM22 complex. The residues likely involved in these interfaces are shown as sticks. The coloring scheme is the same as that in **a**. **c** Electron density map of the area including the LGI1 LRR–ADAM22 and LGI1 LRR–LGI1 EPTP interfaces and linker between LRR and EPTP within LGI1 ($2F_o$–$F_c$ map, contoured at 1.0 $\sigma$ level)

containing protein complexes, showing the interactions with ADAM22, ADAM23, and PSD-95 (Fig. 6a). Quantitative Western blotting showed that the LGI1$^{R474Q}$ binding to ADAM23 was intact, whereas its binding to ADAM22 was reduced as compared with the LGI1$^{WT}$ binding ($42.8 \pm 0.6\%$ reduction) (Fig. 6b, c). However, the partially reduced LGI1$^{R474Q}$ binding to ADAM22 is not sufficient to cause the lethality of $Lgi1^{-/-;R474Q}$ mutant mice, as ADAM22 heterozygous knockout mice do not show any epileptic phenotypes (with 50% of the LGI1–ADAM22 interaction)[39]. We then asked if the LGI1–LGI1 interaction is affected by the R474Q mutation in the brain. When ADAM23 was immunoprecipitated from brain extracts of the wild-type mouse brain, ADAM22 was co-immunoprecipitated completely in an LGI1-dependent manner, indicating that ADAM22 and ADAM23 occur in a tripartite protein complex together with LGI1[34] (Fig. 6d, e; $Lgi1^{+/+}$ versus $Lgi1^{-/-}$). The tripartite complex formation was restored by the reexpression of LGI1$^{WT}$ ($Lgi1^{-/-;WT}$). Importantly, the co-immunoprecipitation of ADAM22 with ADAM23 was robustly reduced in $Lgi1^{-/-;R474Q}$ mouse brain ($79.9 \pm 3.6\%$ reduction) as compared with that in the $Lgi1^{-/-;WT}$ mouse brain (Fig. 6d, e). Given that LGI1$^{R474Q}$ has the intact

binding ability to ADAM23 and partially reduced binding to ADAM22 (less than 50% reduction) (Fig. 6b, c), the LGI1–LGI1 interaction is primarily reduced in the $Lgi1^{-/-;R474Q}$ mouse brain (estimated to be ~35% of that in the $Lgi1^{-/-;WT}$ mouse brain). Reciprocally, co-immunoprecipitation of ADAM23 with ADAM22 was heavily reduced in $Lgi1^{-/-;R474Q}$ mouse brain (Fig. 6f, g). Thus, LGI1–ADAM22 and LGI1–ADAM23 are assembled into higher-order heteromers (at least, heterotetramers) in vivo and the disruption of the inter-LGI1 interactions causes epilepsy in an ADLTE mouse model (Fig. 6h).

## Discussion

Mammalian LGI family consists of four members (LGI1–LGI4). Mutations of LGI family members have been reported to link to various neurological disorders. Truncation of LGI2, which causes its defect in secretion, is associated with focal-onset epilepsy in the *Lagotto Romagnolo* canine breed[47]. LGI4 has been identified as a causative gene of *claw paw* mouse that is a spontaneously arising mutant mouse and displays limb posture abnormalities and peripheral nerve hypomyelination[48]. LGI2 and LGI4 have

also been reported as ligands of ADAM22 in addition to LGI1[47,49]. Consistently, most of the ADAM22-interacting residues of LGI1 are conserved in both LGI2 and LGI4 (Supplementary Fig. 5). Although a pathophysiological function of LGI3 has currently been unclear and the neuronal expression of LGI3

cannot rescue the LGI1 knockout mouse[34], LGI3 also shares most of the ADAM22-interacting residues of LGI1, suggesting its potential binding to ADAM22 family proteins. The Glu–Arg pair for the LGI1–LGI1 interaction is also conserved in LGI2–LGI4 (Supplementary Fig. 5). The LGI family-mediated bridging of

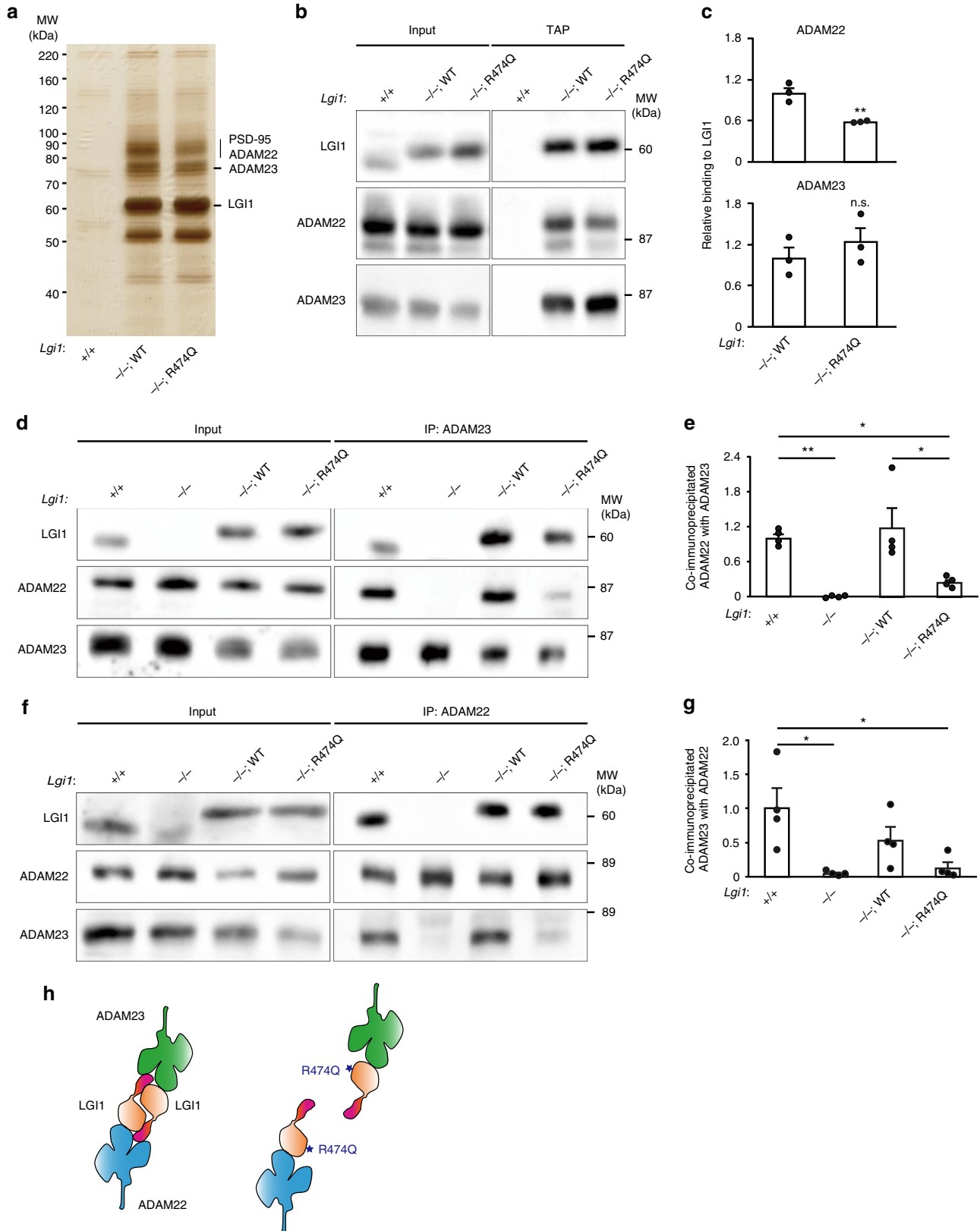

ADAM22 family proteins might play divergent roles in neuronal functions in different cellular contexts.

In this study, we establish a structural basis of LGI1–ADAM22 ligand–receptor complex, which can form the 2:2 LGI1–ADAM22 heterotetramer. On the other hand, the SEC-MALS and cryo-EM analyses of the LGI1$^{R470A}$–ADAM22 ECD complex also suggested the presence of the 3:3 heterohexamer in solution. Additionally, the small angle X-ray scattering (SAXS) curve measured in our SEC-SAXS experiment of the LGI1$^{R470A}$–ADAM22 ECD complex could not be perfectly fitted to the theoretical scattering curve calculated from the 2:2 complex (Supplementary Fig. 6a). It seems feasible that three LGI1 molecules can bridge three ADAM22 molecules as shown in Supplementary Fig. 6b, c. The cryo-EM 2D class averages corresponding to the 3:3 complex exhibited a pseudo C3 symmetry, which allowed us to build a 3:3 model (Supplementary Figs. 4 and 6b, c). This pseudo C3-symmetric 3:3 model could be computationally fitted to the SAXS curve of the LGI1$^{R470A}$–ADAM22 complex (Supplementary Fig. 6b), supporting the presence of the 3:3 complex. This finding may raise the possibility that the switching between the 2:2 and 3:3 assembly modes of the LGI1–ADAM22 complex could be associated with the synaptic function. In both assemblies, the important point is that LGI1 bridges two ADAM22 molecules with the LRR–EPTP interaction between the two adjacent LGI1 molecules.

Previous studies have supported that there are interdependent interactions between postsynaptic ADAM22 and presynaptic ADAM23 involving LGI1 at least in the molecular layer of the dentate gyrus (DG), representing tripartite trans-synaptic complexes. Specifically, the in vivo LGI1-associated protein complex includes both postsynaptic (PSD-95, PSD-93, and SAP97) and presynaptic (CASK, Lin7, and Kv1) proteins[34]. Furthermore, ADAM23 protein in the outer/middle molecular layers of the DG is apparently reduced in ADAM22 and LGI1 knockout mice[9]. ADAM23 protein localized in the DG represents totally presynaptic one derived from the entorhinal cortex, because there is no expression of ADAM23 mRNA in dentate granule cells[32,50]. mRNAs of ADAM22 and LGI1 are highly expressed in dentate granule cells and the corresponding proteins are enriched in the dentate molecular layer[9,32]. In the present structure of the 2:2 LGI1–ADAM22 complex, the C-terminals of the two ADAM22 ECD molecules are oriented in the opposite directions (Fig. 5a) and the length along the longest axis is about 190 Å, which matches the height of the synaptic cleft. These two structural features support the idea that the present structure of the heterotetrameric LGI1–ADAM22 complex reflects the trans-synaptic linkage mediated by the tripartite trans-synaptic complex of ADAM23–LGI1–ADAM22. In the 3:3 LGI1–ADAM22 assembly model, the C-terminals of two of the three ADAM22 ECD molecules are also oriented in the opposite directions and the length along the longest axis is about 160 Å. Nevertheless, we cannot exclude the possibility that a cis-interaction between LGI1 and ADAM22/ADAM23 on the same pre- or postsynaptic membrane occurs under some circumstances in a manner similar to the 2:2 or 3:3 assembly presented in this study.

Given the essential role of LGI1, ADAM22, and ADAM23 in epileptogenesis, LGI1–ADAM22 subfamily tetramers or hexamers may have unique and distinct functions from other numerous trans-synaptic cell adhesion molecules such as neurexin–neuroligin[51]. Recent super-resolution imaging revealed a trans-synaptic nanocolumn that aligns nanometer-scale synaptic subregions, presynaptic RIM-containing nanodomains, and postsynaptic PSD-95-organizing nanodomains, for precise synaptic transmission[52]. Because ADAM22 directly binds to the third PDZ domain of PSD-95, LGI1–ADAM22 subfamily tetramers may participate in the trans-synaptic nanocolumn formation through unknown presynaptic partners. Alternatively, LGI1–ADAM22 tetramers may stabilize the PSD-95 platform as an extracellular scaffold and thereby activate binding activities of PSD-95 at the first/second PDZ domains to AMPA receptor/TARP, NMDA receptor, and Kv1 channels. Consistently, loss of LGI1 and ADAM22 reduces AMPA receptor currents[34,35] and Kv1 expression[34,53]. Future analysis will need to clarify the mode of action of LGI1–ADAM22–PSD-95 supramolecular complex. Lastly, this study proposes that LGI1–ADAM22 represents an intriguing therapeutic target for epilepsy and other neurological disorders. Our present structure might facilitate the development of anti-epilepsy drugs by serving as a useful platform for structure-based design.

## Methods

**Antibodies.** The following commercially available antibodies were used: rabbit polyclonal antibodies to LGI1 (Abcam, ab30868, 1:100 for Western blotting) and ADAM23 (Abcam, ab28302, 1:200); mouse monoclonal antibodies to ADAM22 (NeuroMab, 75-083, 1:250 and 10 µg for immunoprecipitation), FLAG (Sigma-Aldrich, F3165, 1:1000), and β-catenin (BD Biosciences, 610153, 1:500). Rabbit polyclonal antibodies to ADAM22 and ADAM23 were raised against GST-ADAM22 (mouse, residues 858–898, 1:1000) and GST-ADAM23 (mouse, residues 815–829, 10 µg for immunoprecipitation), respectively[9,34].

**Cloning and plasmid constructions.** The cDNA of human LGI1 (NM_005097) was purchased from Thermo Scientific (clone ID: 4811956). The cDNA of human ADAM22 (same sequence as in NM_021723 except the c.242C>G, p.Pro81Arg polymorphism in the Pro domain) was kindly provided by Dr. Toshitaka Kawarai (Tokushima University Graduate School). The cDNA of human ADAM23 (AB009672 (https://www.ncbi.nlm.nih.gov/nuccore/AB009672)) was obtained from Dr. Koji Sagane (Eisai Company)[54]. For pull-down assays, the cDNA of human LGI1 with or without His$_6$ tag at the 3′ end together with 3′UTR and the cDNAs encoding the ECDs of human ADAM22 (residues 35–729) and ADAM23 (residues 61–790) with a FLAG tag at the 3′ end were subcloned into cytomegalovirus promoter-driven expression vectors. To obtain ADAM22 ECD-FLAG and ADAM23 ECD-FLAG as soluble forms, Igκ signal peptide was used instead of authentic signal peptide sequences. Indicated mutations of LGI1, ADAM22, and ADAM23 were introduced by site-directed mutagenesis. For Fig. 3c, the cDNAs encoding human LGI1$^{WT}$ and LGI1$^{R474Q}$ (residues 37–557) tagged with His$_6$ were subcloned into pEBMulti-Neo. All PCR products were analyzed by DNA sequencing (Functional Genomics Facility, NIBB). All primer sequences are shown in Supplementary Data 1.

**Protein preparations and crystallization.** The genes encoding human LGI1 proteins (LRR, residues 37–223; EPTP, residues 224–557; full length, residues 37–557) with the N-terminal Igκ signal sequence and C-terminal His$_6$ tag and ADAM22 ECD including the N-terminal prosequence (residues 1–729) with or without the C-terminal His$_6$ tag were cloned into the pEBMulti-Neo vector (Wako chemicals). All these proteins were transiently expressed in Expi293F cells (Thermo

**Fig. 6** Pathogenic mechanism of a secretion-competent ADLTE mutation, LGI1$^{R474Q}$. **a–c** Tandem-affinity purification (TAP) of LGI1$^{WT}$ and LGI1$^{R474Q}$ tagged with FLAG and His$_6$ from the indicated mouse brain extracts. Shown are the silver staining of TAP eluates (**a**) and Western blots of input (left) and TAP eluates (right) with indicated antibodies (**b**). Quantification of the amount of co-purified ADAM22 and ADAM23 with tagged LGI1 is shown in the graph (**c**). Known co-purified proteins were indicated (**a**). **P < 0.01; n.s. not significant; $n = 3$ independent experiments (**c**). Two-tailed Student's t test was used. **d, e** Immunoprecipitation (IP) of ADAM23 from the indicated mouse brain extracts. Shown are Western blots of input (left) and IP (right) samples with indicated antibodies (**d**). Quantification of the amount of ADAM22 co-immunoprecipitated with ADAM23 is shown in the graph (**e**). **f, g** IP of ADAM22 from the indicated mouse brain extracts. **P < 0.01; *P < 0.05; $n = 4$ independent experiments. One-way ANOVA followed by post hoc Tukey's test was used (**e, g**). Results are shown as mean ± s.e. **h** Model of tripartite complex comprising ADAM22–LGI1–ADAM23 at 1:2:1 stoichiometry. Two heterodimers, LGI1–ADAM22 and LGI1–ADAM23, are arranged in the LGI1-mediated head-to-head configuration to form the tetrameric complex (left). The R474Q mutation (asterisk) in LGI1 disrupts the LGI1–LGI1 interaction (right)

Fisher Scientific). For the preparation of the LGI1 EPTP–ADAM22 complex, the C-terminally His$_6$-tagged LGI1 EPTP was co-expressed with the C-terminally His$_6$-tagged ADAM22. For the preparation of the LGI1–ADAM22 complex, the C-terminally His$_6$-tagged LGI1 was co-expressed with the non-tagged ADAM22. The culture media were loaded onto a Ni-NTA (Qiagen) column pre-equilibrated with 20 mM Tris-HCl (pH 8.0) containing 300 mM NaCl. After the column was washed with 20 mM Tris-HCl (pH 8.0) containing 300 mM NaCl and 25 mM imidazole, the proteins were eluted with 20 mM Tris-HCl (pH 8.0) containing 300 mM NaCl and 250 mM imidazole. The eluted proteins were further purified by SEC using Superdex200 (GE healthcare) with 20 mM Tris-HCl (pH 7.5) buffer containing 150 mM NaCl. The purified proteins were concentrated to 1–2 g L$^{-1}$ using an Amicon Ultra-4 30,000 MWCO filter (Millipore), flash-frozen in liquid N$_2$, and stored at −80 °C until use.

Crystals were grown by the sitting-drop vapor diffusion method at 20 °C by mixing a protein solution and a crystallization solution in a 1:1 (v/v) ratio. The formulations of the crystallization solutions were as follows: 20% PEG3350, 0.2 M magnesium nitrate, 3% methanol for the LGI1 EPTP–ADAM22 ECD complex; 20% PEG4000, 0.2 M ammonium acetate, 0.1 M tri-sodium citrate (pH 5.5) for LGI1 LRR; 10% PEG8000, 0.1 M zinc acetate, 0.1 M MES-NaOH (pH 6.0) for the LGI1–ADAM22 ECD complex. The crystals were soaked in the crystallization solutions supplemented with 30% ethylene glycol and then flash-frozen in liquid N$_2$.

**Structure determination.** Diffraction data sets were collected at 100 K at BL41XU in SPring-8 (Hyogo, Japan) and processed with HKL2000[55] and the CCP4 program suite[56]. The LGI1 LRR structure was determined by molecular replacement using the program Balbes[57], which selected the Slit2 D4 LRR structure (PDB 2WFH) as the optimal reference model. The structure of the LGI1 EPTP–ADAM22 complex was determined by molecular replacement using the program Molrep[58]. The apo ADAM22 ECD structure (PDB 3G5C) was used as the search model. The atomic model of the entire LGI1 EPTP domain could be obtained after iterative cycles of model building and structure refinement. Information from the 3D structure prediction of LGI1 EPTP[26] was useful for interpretation of the electron density map at the initial stage of the model building. The structure of the LGI1–ADAM22 complex was also determined by molecular replacement using the program Molrep[58]. The structures of LGI1 LRR and the LGI1 EPTP–ADAM22 ECD complex were used as the search models. The programs Coot[59] and Phenix[60] were used for the model building and structure refinement, respectively. Data collection and refinement statistics are shown in Table 1.

**Pull-down assay.** Expression vectors for ADAM22 ECD-FLAG, ADAM23 ECD-FLAG, LGI1, or LGI1-His$_6$ (see 'Cloning and plasmid constructions') were transfected into HEK293T cells, which were confirmed as mycoplasma-free. At 24 h after transfection, the cells were washed with serum-free DMEM and cultured for an additional 24 h under serum-free conditions. Each culture medium was collected and mixed for 1 h at 4 °C. LGI1-His$_6$ and ADAM22/23 ECD-FLAG were then purified by Ni-NTA agarose beads (Qiagen) and anti-FLAG M2 agarose beads (Sigma-Aldrich), respectively. The purified proteins were separated by SDS-PAGE and subjected to Western blotting with anti-FLAG and anti-LGI1 antibodies. For quantitative Western blotting, chemical luminescent signal was detected with a cooled CCD camera (Light-Capture II; ATTO) or the FUSION Solo system (Vilber-Lourmat). The band intensities were analyzed with CS analyzer 3.0 software (ATTO) or the FUSION Solo system. Uncropped images of blots are shown in Supplementary Fig. 7.

**SEC-MALS.** Samples (1 g L$^{-1}$) were applied onto an ENrich SEC 650 (10 × 300 mm) column (Bio-Rad) with 20 mM Tris-HCl (pH 7.5) buffer containing 150, 300, or 500 mM NaCl. The MALS data were collected by a DAWN HELEOS 8+ detector (Wyatt Technology) with RF-20A UV detector (Shimadzu) and analyzed by the program ASTRA (Wyatt Technology).

**Cryo-EM.** A 4.0-μL of purified sample of the LGI1$^{R470A}$–ADAM22 ECD complex (95 ng μL$^{-1}$) was applied to a glow-discharged holey carbon grid (R1.2/1.3, Quantiofoil), and the grid was plunge-frozen into liquid ethane by using a semi-automated vitrification device (Vitrobot Mark IV, FEI) with 3-s blotting with 0 blot offset in 100% humidity at 4 °C. Data acquisition was performed by using 200-kV field emission cryo-EM (Tecnai Arctica, FEI) at 23,500-fold nominal magnification with a Gatan K2 summit direct electron detector under low-dose condition using the data acquisition software Serial EM[61]. All the data were collected as a movie with 36 subframes of 1.4 e$^{-1}$ per Å$^2$ in super-resolution mode with a total electron dose of 50 e$^{-1}$ per Å$^2$ at a pixel size of 0.785 Å per pixel. The defocus range of the data set was set to a range of −1.5 − −3.5 μm.

Movie processing was performed using the software MotionCor2[62] and CTFFIND4[63]. After the particle picking performed with Gautomatch (http://www.mrc-lmb.cam.ac.uk/kzhang/) for the particles with 200 Å diameter, image processing was performed with RELION2[64,65]. Particles were extracted with relatively larger box size, 192 × 192 pixels to have more than one 2:2 LGI1$^{R470A}$–ADAM22 ECD complex in it. An initial 2D classification was performed with 280 Å diameter mask, which is large enough to accommodate the

oligomer particles larger than the dimer particles. The second 2D classification was performed to roughly assign the classes to monomers, dimers, and trimers. Several steps for 2D classification were performed to extract monomers from the dimer classes or dimers from the trimer classes. The final classification for the monomer, which corresponds to the 1:1 LGI1$^{R470A}$–ADAM22 ECD complex, was performed to 50 classes with the mask of 150 Å. Thirty-nine classes were then selected as the 1:1 LGI1$^{R470A}$–ADAM22 ECD complex (46,153 particles). The final classification for the dimer, which corresponds to the 2:2 LGI1$^{R470A}$–ADAM22 ECD complex, was performed to 50 classes with the mask of 280 Å to select only dimer classes. Twenty-four classes were then selected as the 2:2 LGI1$^{R470A}$–ADAM22 ECD complex (21,163 particles). The final classification for the trimer, which corresponds to the 3:3 LGI1$^{R470A}$–ADAM22 ECD complex, was performed to 40 classes with the mask of 280 Å to select only trimer classes. Thirteen classes were then selected as the 3:3 LGI1$^{R470A}$–ADAM22 ECD complex (3780 particles).

**SEC-SAXS.** SEC-SAXS data were collected on beamline BL45XU at SPring-8 (Hyogo, Japan). The LGI1–ADAM22 complex at two different concentrations (5.7 g L$^{-1}$ or 8.2 g L$^{-1}$) was applied onto an ENrich SEC 650 (10 × 300 mm) column (Bio-Rad) with 20 mM Tris-HCl (pH 7.5) buffer containing 150 mM or 500 mM NaCl. Scattering intensities were measured at 293 K on PILATUS 3 × 2 M detector with sample to detector distance of 3.5 m. Data collection and structural parameters are listed in Supplementary Table 2. Four scattering data (0.25 s radiation for each measurement) at elution top peak were averaged by the program PRIMUS[66] and used for the subsequent analysis. The molecular modeling of the 3:3 LGI1–ADAM22 assembly and the subsequent rigid body fitting to the experimental scattering curve were performed using the program SASREF[67]. In this fitting, the LGI1 EPTP–ADAM22 complex was treated as a single rigid body, while LGI1 LRR was treated as another rigid body. In addition, the Cα–Cα distance restraints were applied on the basis of the crystal structure as follows: 4 Å distance between Ile222 and Ile223 (the two-residue linker between LGI1 LRR and EPTP) and 12 Å distance between Arg474 of one LGI1 and Glu123 of the other LGI1. A simulated annealing protocol was employed to minimize the difference between the experimental and theoretical scattering curves. This fitting was performed using the program CRYSOL[68].

**Generation of transgenic mice.** All of the animal studies were reviewed and approved by the Institutional Animal Care and Use Committee of National Institutes of Natural Sciences and were performed in accordance with its guidelines concerning the care and handling of experimental animals. Lgi1 KO mouse strain was previously established[34]. Briefly, embryonic stem cell clones with the targeted Lgi1 locus were injected into C57BL/6 blastocysts. The chimeras were crossed with C57BL/6 mice for germ line transmission. Tandemly tagged Lgi1$^{R407C}$ and Lgi1$^{R474Q}$ transgenic mice (with FLAG and His$_6$ tags) were generated by DNA injection into fertilized embryos as for tandemly tagged Lgi1$^{WT}$ transgenic mice[34]. Briefly, the cDNA of Lgi1$^{R407C}$ or Lgi1$^{R474Q}$ with FLAG and His$_6$ tags was sub-cloned downstream of the Thy1 promoter[69]. Obtained transgenic founders were crossed with C57BL/6 mice and genotyping was performed using PCR primers shown in Supplementary Data 1. For the rescue experiment, the Lgi1$^{+/-}$ mouse was crossbred with an Lgi1$^{WT}$, Lgi1$^{R407C}$, or Lgi1$^{R474Q}$ transgenic mouse. Obtained Lgi1$^{+/-;WT}$, Lgi1$^{+/-;R407C}$, or Lgi1$^{+/-;R474Q}$ was crossed with Lgi1$^{+/-}$ to obtain Lgi1$^{-/-;WT}$, Lgi1$^{-/-;R407C}$, or Lgi1$^{-/-;R474Q}$. Slightly prolonged lifetime of Lgi1$^{-/-;R474Q}$ mice as compared with Lgi1 null mice (Fig. 3d) is probably due to the still remaining functional tripartite complexes (Fig. 6e, g).

**Tandem-affinity purification and immunoprecipitation.** Brains from Lgi1$^{-/-;WT}$ and Lgi1$^{-/-;R474Q}$ mice (for Fig. 6) or from Lgi1$^{+/-;WT}$ and Lgi1$^{+/-;R407C}$ mice (for Supplementary Fig. 2c) were homogenized and expressed LGI1$^{WT}$, LGI1$^{R407C}$, or LGI1$^{R474Q}$-FLAG-His$_6$ was purified. Briefly, homogenates were spun at 20,000 g for 1 h and pellets (plasma-membrane fractions) were solubilized with 20 mM Tris-HCl (pH 7.5) buffer containing 1.3% Triton X-100, 2 mM EDTA, and 50 μg mL$^{-1}$ phenylmethylsulfonyl fluoride (PMSF). After centrifugation at 100,000×g for 1 h, the supernatant was incubated with anti-FLAG M2 agarose. The first eluate was obtained with 20 mM Tris-HCl (pH 7.5) buffer containing 100 mM NaCl, 1% Triton X-100, 50 μg mL$^{-1}$ PMSF, 20 mM imidazole, and 0.25 g L$^{-1}$ FLAG peptide. The eluate was mixed with Ni-NTA agarose and the second eluate was obtained with 20 mM Tris-HCl (pH 7.5) buffer containing 100 mM NaCl, 1% Triton X-100, 50 μg mL$^{-1}$ PMSF, and 250 mM imidazole. The purified proteins were separated by SDS-PAGE, subjected to silver staining or Western blotting. For immunoprecipitation of ADAM22 or ADAM23 from mouse brain extracts (postnatal days 17–18), the plasma-membrane fractions were solubilized with 20 mM Tris-HCl (pH 8.0) buffer containing 1.3% Triton X-100, 2 mM EDTA, and 50 μg mL$^{-1}$ PMSF. After centrifugation at 100,000×g for 1 h, the supernatant was precleared and incubated with 10 μg of anti-ADAM22 or anti-ADAM23 antibodies for 1 h at 4 °C. The immunecomplex was then precipitated with Protein A Sepharose and eluted with SDS-PAGE sample buffer. Eluates were separated by SDS-PAGE and subjected to Western blotting. Uncropped images of gels and blots are shown in Supplementary Fig. 7.

**Statistical analysis**. To perform statistical analysis, at least three independent experiments, tissue samples, or mice were included in the analyses. No statistical method was used to determine the sample size. No data were excluded. There was no randomization of mice or samples before analysis, and the mice used in this study were selected based purely on availability. For paired sample comparisons, two-tailed Student's *t* test was used; and for multiple test subjects, one-way ANOVA with appropriate post hoc tests (as indicated in the figure legends) was used. Survival curves of LGI1 mutant mice were analyzed by Kaplan–Meier survival estimate using a log-rank test for curve comparisons (Fig. 3d). Statistical analysis was performed with Ekuseru-Toukei 2012 software (SSRI). Results are shown as mean ± standard error.

**Code availability**. The code of the program DataProcess for SAXS measurement at BL45XU of SPring-8 can be downloaded from 'https://beamline.harima.riken.jp/bl45xu/DataProcess/'.

**Data availability**. The coordinates and structure factors of LGI1 LRR and the LGI1 EPTP–ADAM22 ECD and LGI1–ADAM22 complexes have been deposited in the Protein Data Bank under the accession codes of 5Y30, 5Y2Z, and 5Y31, respectively. Primer sequences used in this study are listed in Supplementary Data 1. Other data are available from the corresponding authors upon reasonable request.

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

## Acknowledgements

We thank Drs. Toshitaka Kawarai (Tokushima University Graduate School) and Koji Sagane (Eisai Company) for providing the cDNAs of human *ADAM22* and *ADAM23*, respectively. We thank Ms. Tomomi Uchikubo-Kamo for the initial assessment by using negative stain electron microscopy and cryo-specimen preparation of the LGI1^R470A–ADAM22 ECD complex. We thank the beamline staff of the biological crystallography and SAXS beamlines of Photon Factory (Tsukuba, Japan) and BL41XU and BL45XU of SPring-8 (Hyogo, Japan) for technical help during data collection. This work was supported by Platform Project for Supporting Drug Discovery and Life Science Research (Basis for Supporting Innovative Drug Discovery and Life Science Research (BINDS)) from Japan Agency for Medical Research and Development (AMED). This work was supported by JSPS/MEXT KAKENHI (16H04749 to A.Y., 17J07958 to Y.M., 17K14969 to N.Y., 17H05897 to H.S., 15H04279 to Y.F., and 17H03678, 17H05709 to M. F.), NCNP (27-7 to Y.F.), The Japan Epilepsy Research and Kato Memorial Bioscience Foundations (to Y.F.), Takeda Science Foundation (to Y.F. and M.F.), The Hori Sciences and Arts Foundation (to M.F.), and JST CREST (JPMJCR12M5 to S.F.).

## Author contributions

A.Y. performed structure determination, SEC-MALS and SEC-SAXS, and wrote the paper. Y.M. performed the pull-down assays. Y.M. and N.Y. characterized LGI1 mutant mice. T.G., M.Sa., and M.H. generated transgenic mice. Y.S., S.G.-I., and S.F. assisted with structure determination. A.M. prepared the samples for crystallography and SEC-MALS. H.S. performed cryo-EM with the aid of M.Sh. and analyzed the data. All authors discussed the results and commented on the manuscript. S.F., M.F., and Y.F. supervised the work, designed the experiments, and wrote the paper.

## Additional information

**Competing interests:** The authors declare no competing interests.

