## [Peer Review File · Nature Communications]

Reviewers' comments:

Reviewer #1 (Remarks to the Author):

In this interesting manuscript, the authors resolved new crystal structures of LGI1 LRR, LGI1 EPTP/ADAM22 ECD complex and LGI1 full/ADAM22 ECD complex revealing their interaction interfaces, and found that LGI1 and ADAM22 can form the 2:2 heterotetrameric complex. In addition, based on their structural and biochemical information, the authors proposed a model of tripartite trans-synaptic complex comprising ADAM22/two LGI1/ADAM23 and a pathogenic mechanism of epilepsy caused by LGI1 mutations. I believe that this paper is generally suitable for Nature Communications but have several comments. Some of them will have to be addressed by showing more experimental data.

1. The data in Fig.4a indicates that LGI1 full/ADAM22 ECD complex can form 2:2 heterotetrameric assembly only at the high salt condition. In general, determining the exact size of protein complex having an elongated shape such as heterotetrameric LGI1 full/ADAM22 ECD complex, with SEC-MALS analysis could be inaccurate. Interestingly, the LGI1 full/ADAM22 ECD complex for the crystallization was resolved in the buffer containing 20mM Tris-Cl pH7.5 and 150mM NaCl, suggesting the protein complex used in crystallization would be 3:3 assembly (according to SEC-MALS data). However, the stoichiometry of LGI1 full/ADAM22 ECD complex in their crystal structure was revealed as 2:2. Therefore, EM (negative staining TEM and 2D-Class average) or other techniques should be used to better define the stoichiometry of LGI1 full/ADAM22 ECD complex in solution.

Moreover, the stoichiometry mismatch in solution (3:3) and in the crystal (2:2) raised a question whether the author can find any meaningful interactions between adjacent LGI1 full/ADAM22 ECD complexes in the crystal lattice to form 6:6 complex. In other words, is it possible that LGI1 binding can induce the higher-order clustering of tripartite trans-synaptic complex comprising ADAM22/two LGI1/ADAM23, like LAR-RPTP-mediated trans-synaptic complex (Won, S.Y., et al., LAR-RPTP Clustering Is Modulated by Competitive Binding between Synaptic Adhesion Partners and Heparan Sulfate. *Frontiers in Molecular Neuroscience*, 2017. 10(327))?

2. Even though low-resolution structure (LGI1 full/ADAM22 ECD complex) and high B factor was obtained (7.12Å and 450Å²), the authors described interactions of LGI1 LRR with ADAM22 ECD and LGI1* EPTP in detail (Fig. 5b). The electron density map of interaction interfaces and of linkers between LGI1 LRR and LGI1 EPTP should be provided. In addition, to convince the docking of high resolution structures (LGI1 LRR and LGI1 EPTP/ADAM22 ECD complex) into the low resolution density map, please show that the mutation of residues on LGI1 LRR domain (R76, H116 or E123) and/or on ADAM22 ECD (E446) can disturb the 2:2 heterotetrameric complex formation, at least in vitro like Fig.4a (LGI1 R474Q on LGI1 EPTP).

3. Fig. 6D. Please, show the results of western blot after immunoprecipitation of ADAM22.

4. The author proposed that LGI1 can mediate synapse-adhesion in a trans-configuration by binding to ADAM22 and ADAM23. Although Fig6 indicates LGI1 can bridge between ADAM22 and ADAM23, it is unclear whether these complex formations are in a trans-manner. Moreover, it is likely that similar interactions between two LGI1 and two ADAM22 on the same synaptic membrane can be made as well. Please, comment on these issues in the Result and/or Discussion section.

Reviewer #2:

LGI1 is a secreted neurotransmitter essential for the regulation of synaptic transmission and therefore the maintenance of brain excitability balance. Missense mutations in LGI1 have been reported in autosomal dominant lateral temporal lobe epilepsy (ADLTE). Most of them lead to secretion defects while others are assumed to have an impact on the transsynaptic ADAM22/23 complex interaction.

In this study, authors achieve a thorough characterization of the structural organization of LGI1-ADAM22 ligand-receptor complex. They further investigate the conformational changes of selected missense mutations to test their model and shed light over potential pathomechanisms underlying LGI1-related epilepsy. The structure of LGI1-ADAM22 complex and its arrangement into heterotetramers at the synapse also brings valuable information to test for the pathogenicity prediction of missense mutations.

The paper is overall very well-written, and presented. This is the first crystal structure of LGI1 and ADAM22. Previous findings from the group and others have shown that the disruption of LGI1-ADAM interaction is implicated in the pathogenic mechanism of ADEAF. This study brings the structural proof of this mechanism.

Please find below suggestions to improve the manuscript:

Abstract:

- Authors should remove the term "refractory" since ADLTE is mostly pharmacosensitive.
- The term "monogenic mutations" is not appropriate.
- The term "epileptogenic mechanism" should be replaced by pathogenic mechanism since this study address the effect of mutations on the structure of LGI1 and formation with the ADAM22 complex, rather than epileptogenesis processes.

Results

- Authors conclude that R407C is not pathogenic based on the fact that expression of this mutant could rescue the mice epileptic phenotype. This is also consistent with the fact that this variant is found in 5 GnomAD controls. This latter point could be mentioned as well.
- The section concerning the ADLTE-associated variants would benefit from a table summarizing their consequences on the structure of the protein, associated to the reference publishing the variants.

Comments from Reviewer #1:

1. The data in Fig.4a indicates that LGI1 full/ADAM22 ECD complex can form 2:2 heterotetrameric assembly only at the high salt condition. In general, determining the exact size of protein complex having an elongated shape such as heterotetrameric LGI1 full/ADAM22 ECD complex, with SEC-MALS analysis could be inaccurate. Interestingly, the LGI1 full/ADAM22 ECD complex for the crystallization was resolved in the buffer containing 20mM Tris-Cl pH7.5 and 150mM NaCl, suggesting the protein complex used in crystallization would be 3:3 assembly (according to SEC-MALS data). However, the stoichiometry of LGI1 full/ADAM22 ECD complex in their crystal structure was revealed as 2:2. Therefore, EM (negative staining TEM and 2D-Class average) or other techniques should be used to better define the stoichiometry of LGI1 full/ADAM22 ECD complex in solution.

Theoretically, SEC-MALS can determine the absolute molar mass of the measured sample, independently of its shape. However, practically, the separation of the sample by SEC is not always perfect, and therefore the determined molar mass sometimes slightly differs from the calculated molar mass. In addition, the heterogeneity of the N-linked sugar chains makes it difficult to determine the accurate molar mass in the case of extracellular proteins.

In the LGI1–ADAM22 ECD complex, our SEC-MALS analysis suggested the 2:2 and 3:3 assemblies. The crystal structure of the complex illustrated the 2:2 assembly. We also examined the purified LGI1–ADAM22 ECD complex sample by cryo-electron microscopy (cryo-EM) and small angle X-ray scattering (SAXS) analysis, according to this comment. The cryo-EM analysis based on 2D class average supported the presence of the 2:2 complex in solution but also suggested that a small fraction of the sample forms the 3:3 complex exhibiting a pseudo C_3 symmetry. A model of the 3:3 complex deduced from this pseudo C_3 symmetry and the LGI1–ADAM22 interactions found in the 2:2 complex could be computationally fitted to the SAXS curve obtained from the SEC-SAXS analysis of the LGI1–ADAM22 ECD complex sample. The majority of the higher-order assembly of the LGI1–ADAM22 ECD complex is likely to be the 2:2 assembly, whereas the 3:3 complex can also be formed in solution. In both complexes, the important point is that LGI1 bridges two ADAM22 molecules, likely for the *trans*-synaptic connection.

The cryo-EM analysis is mentioned in the third paragraph of the subsection "Dimer-of-dimer assembly of LGI1–ADAM22" (pg. 12; Supplementary Fig. 4) and the second paragraph of the Discussion section (pg. 15-16). The SEC-SAXS analysis is mentioned in the second paragraph of the Discussion section (pg. 15-16; Supplementary Fig. 6).

*Moreover, the stoichiometry mismatch in solution (3:3) and in the crystal (2:2) raised a question whether the author can find any meaningful interactions between adjacent LGI1 full/ADAM22 ECD complexes in the crystal lattice to form 6:6 complex. In other words, is it possible that LGI1 binding can induce the higher-order clustering of tripartite trans-synaptic complex comprising ADAM22/two LGI1/ADAM23, like LAR-RPTP-mediated trans-synaptic complex (Won, S.Y., et al., LAR-RPTP Clustering Is Modulated by Competitive Binding between Synaptic Adhesion Partners and Heparan Sulfate. *Frontiers in Molecular Neuroscience*, 2017. 10(327))?*

Our crystallography, SEC-MALS, cryo-EM and SEC-SAXS analyses of the LGI1-ADAM22 complex indicate that it can form both the 2:2 and 3:3 complexes in solution. These two assembly states could be equilibrated. We therefore speculate the possibility that the switching between the 2:2 and 3:3 complexes could be associated with the synaptic function. This point is mentioned in the second paragraph of the Discussion section (pg. 15-16).

Regarding a possibility that LGI1 binding can induce the higher-order lateral clustering like other *trans*-synaptic adhesion complexes such as Ila RPTP-IL1RAP1 (Won, S.Y., et al., *Front. Mol. Neurosci.* 10, 327 (2017)), Ila RPTP-Slitrk1 (Um, J.W. et al., *Nat. Commun.* 5, 5423 (2014)) and β -neurexin-neurologin (Tanaka, H. et al., *Cell Rep.* 2,101-110 (2012)), we could find no obvious meaningful interactions between the adjacent LGI1-ADAM22 ECD complexes in the crystal lattice. Alternatively, the clustering of ADAM22 might be induced by its intracellular binding partners, 14-3-3 and PSD-95, each of which can dimerize. Subsequently, LGI1 binds to ADAM22 and ADAM23, mediates *trans*-synaptic linkages, and stabilizes them at the synaptic membrane. In fact, ADAM22 and ADAM23 both are greatly delocalized from the synaptic fraction in LGI1 knockout mice (Fukata, Y., et al., *PNAS*, 107, 3799-3804 (2010); Yokoi, N. et al., *Nat. Med.*, 21, 19-26 (2015)). There may be extracellular and intracellular clustering mechanisms to cooperatively induce the possible higher-order clustering of the LGI1-ADAM22 family complexes. We will not describe the details of this point (*i.e.*, higher-order clustering of the LGI1-ADAM22 family complex) in the text because we have no sufficient experimental evidence to support it.

2. Even though low-resolution structure (LGI1 full/ADAM22 ECD complex) and high B factor was obtained (7.12Å and 450Å²), the authors described interactions of LGI1 LRR with ADAM22 ECD and LGI1 EPTP in detail (Fig. 5b). The electron density map of interaction interfaces and of linkers between LGI1 LRR and LGI1 EPTP should be provided. In addition, to convince the docking of high resolution structures (LGI1 LRR and LGI1 EPTP/ADAM22 ECD complex) into the low resolution density map, please show that the mutation of residues on LGI1 LRR domain (R76, H116 or E123) and/or on ADAM22 ECD (E446) can disturb the 2:2 heterotetrameric complex formation, at least in vitro like Fig.4a (LGI1 R474Q on LGI1 EPTP).*

The electron density map of the LGI1 LRR–ADAM22 and LGI1 LRR–LGI1 EPTP interfaces and that of the linker between LRR and EPTP within LGI1 were shown in Fig. 5c. We also showed the SEC-MALS data in which the R76A mutation of LGI1 disturbs the higher-order assembly of the LGI1–ADAM22 complex (Supplementary Fig. 3b). This result was described in the last sentence of the subsection “Dimer-of-dimer assembly of LGI1–ADAM22” (pg. 13).

3. Fig. 6D. Please, show the results of western blot after immunoprecipitation of ADAM22.

We performed the immunoprecipitation of ADAM22 and found that ADAM23 was co-immunoprecipitated with ADAM22 in an LGI1-dependent manner as previously reported (Fukata, Y. *et al.*, *PNAS*, 107, 3799-3804 (2010)). Consistently, co-immunoprecipitation of ADAM23 with ADAM22 was heavily reduced in *Lgi1*^{-/-;R474Q} mouse brain. We added this data in the revised version (the second last sentence in the subsection “Disruption of LGI1–LGI1 interaction causes epilepsy” in pg. 14; Fig. 6f, g).

4. The author proposed that LGI1 can mediate synapse-adhesion in a trans-configuration by binding to ADAM22 and ADAM23. Although Fig6 indicates LGI1 can bridge between ADAM22 and ADAM23, it is unclear whether these complex formations are in a trans-manner. Moreover, it is likely that similar interactions between two LGI1 and two ADAM22 on the same synaptic membrane can be made as well. Please, comment on these issues in the Result and/or Discussion section.

The reviewer wondered if the interactions between two LGI1 and two ADAM22 molecules could occur on the same synaptic membrane (*i.e.*, in a *cis*-configuration). We discussed this point in the third paragraph of the Discussion section (pg. 16) as follows:

“Previous studies have supported that there are interdependent interactions between postsynaptic ADAM22 and presynaptic ADAM23 involving LGI1 at least in the molecular layer of the dentate gyrus (DG), representing tripartite *trans*-synaptic complexes. Specifically, the *in vivo* LGI1-associated protein complex includes both postsynaptic (PSD-95, PSD-93 and SAP97) and presynaptic (CASK, Lin7 and Kv1) proteins. Furthermore, ADAM23 protein in the outer/middle molecular layers of the DG is apparently reduced in ADAM22 and LGI1 knockout mice. ADAM23 protein localized in the DG represents totally presynaptic one derived from the entorhinal cortex, because there is no expression of ADAM23 mRNA in dentate granule cell. mRNAs of ADAM22 and LGI1 are highly expressed in dentate granule cells and the corresponding proteins are enriched in the dentate molecular layer. In the present structure of the 2:2 LGI1–ADAM22 complex, the C-terminals of the two ADAM22 ECD molecules are oriented in the opposite directions and the length along the longest axis is about 190 Å, which matches the

height of the synaptic cleft. These two structural features support the idea that the present structure of the heterotetrameric LGI1–ADAM22 complex reflects the *trans*-synaptic linkage mediated by the tripartite *trans*-synaptic complex of ADAM23–LGI1–ADAM22. In the 3:3 LGI1–ADAM22 assembly model, the C-terminals of two of the three ADAM22 ECD molecules are also oriented in the opposite directions and the length along the longest axis is about 160 Å. Nevertheless, we cannot exclude the possibility that a *cis*-interaction between LGI1 and ADAM22/ADAM23 on the same pre- or postsynaptic membrane occurs under some circumstances in a manner similar to the 2:2 or 3:3 assembly presented in this study.”

Comment from Reviewer #2:

Abstract:

- *Authors should remove the term “refractory” since ADLTE is mostly pharmacosensitive.*

The term “refractory” was removed accordingly.

- *The term “monogenic mutations” is not appropriate.*

The term “monogenic mutations” was removed accordingly.

- *The term “epileptogenic mechanism” should be replaced by pathogenic mechanism since this study address the effect of mutations on the structure of LGI1 and formation with the ADAM22 complex, rather than epileptogenesis processes.*

We removed the term “*epileptogenic mechanism*” and rewrote the sentence including it as follows:

“... These studies support the notion that the LGI1–ADAM22 complex functions as the *trans*-synaptic machinery for precise synaptic transmission.”

- *Authors conclude that R407C is not pathogenic based on the fact that expression of this mutant could rescue the mice epileptic phenotype. This is also consistent with the fact that this variant if found in 5 GnomAD controls. This latter point could be mentioned as well.*

The reviewer kindly advised us to mention that R407C variant is found in 5 gnomAD controls. We mentioned this in the revised version as follows (pg. 9, line 9).

“... Furthermore, when the expressed LGI1^{R407C} was purified from the mouse brain, LGI1^{R407C} bound to ADAM22 and ADAM23 as LGI1^{WT} did (Supplementary Fig. 2c). Consistently, LGI1^{R407C} variant was found in 5 gnomAD controls (the Genome Aggregation Database; <http://gnomad.broadinstitute.org/>). We thus conclude that R407C is not a pathogenic mutation for ADLTE.”

- The section concerning the ADLTE-associated variants would benefit from a table summarizing their consequences on the structure of the protein, associated to the reference publishing the variants.

The information of the ADLTE-associated variants (their consequences, positions in the 3D structure, and related references) was summarized as Supplementary Table 1.

REVIEWERS' COMMENTS:

Reviewer #1 (Remarks to the Author):

The authors made excellent additions to the text and fully addressed my comments to strengthen their conclusion. I think that the current paper is acceptable for the publication.